# Effects of *Saccharomyces cerevisiae* Culture on Ruminal Fermentation, Blood Metabolism, and Performance of High-Yield Dairy Cows

**DOI:** 10.3390/ani11082401

**Published:** 2021-08-13

**Authors:** Xiaoge Sun, Yue Wang, Erdan Wang, Shu Zhang, Qianqian Wang, Yan Zhang, Yajing Wang, Zhijun Cao, Hongjian Yang, Wei Wang, Shengli Li

**Affiliations:** 1State Key Laboratory of Animal Nutrition, Beijing Engineering Technology Research Center of Raw Milk Quality and Safety Control, College of Animal Science and Technology, China Agricultural University, No. 2 Yuanmingyuan West Road, Haidian, Beijing 100193, China; xiaogesun@163.com (X.S.); wangerdan@cau.edu.cn (E.W.); zhangshu200666@126.com (S.Z.); qq_221w@163.com (Q.W.); dxzxyan8@163.com (Y.Z.); yajingwang@cau.edu.cn (Y.W.); caozhijun@cau.edu.cn (Z.C.); yang_hongjian@cau.edu.cn (H.Y.); 2Animal Production Systems Group, Wageningen University & Research, 6708 PB Wageningen, The Netherlands; yue3.wang@wur.nl

**Keywords:** *Saccharomyces cerevisiae*, high-yield cows, pH, VFA, inflammatory cytokines

## Abstract

**Simple Summary:**

Nowadays, the lifetime milk production of dairy cows, as well as the fat and protein contents of milk, has reached an unprecedented high. These improvements pose threats and challenges to animal health and welfare due to metabolic stress. The cows, during the high-yielding period, are especially susceptible to metabolic diseases such as digestive alterations, rumen acidosis, and lameness. This study assessed the effects of *Saccharomyces cerevisiae* culture (SC), a food supplement, on ruminal pH, volatile fatty acid (VFA), inflammatory cytokines, and the performance of high-yield dairy cows. The results show that supplementing high-yield lactating cows with the SC of 100 g/d increases milk yield, milk fat content, and milk lactose content, but does not affect protein content. SC supplementation affects overall ruminal VFA concentration and induces a significantly greater ruminal pH. It has the potential to enhance the rumen microbial growth and decrease the inflammation response. Our research suggests that SC supplementation has a positive effect on the productivity and health of dairy cows.

**Abstract:**

High-yield dairy cows with high-concentrate diets are more prone to experiencing health problems associated with rumen microbial imbalance. This study assessed the effects of *Saccharomyces cerevisiae* culture (SC), a food supplement, on ruminal pH, volatile fatty acid (VFA), inflammatory cytokines, and performance of high-yield dairy cows. Forty Holstein cows with similar characteristics (e.g., milk yield, days of milk, and parity) were randomly divided into two groups: an experimental group fed the basal ration supplemented with the SC of 100 g of SC per cow per day (hour, SC group), and a control group fed the same basal ration diet without SC (i.e., CON group). On average, the supplementation of SC started at 73 days of lactation. The experimental period lasted approximately 70 days (from 18 January to 27 March 2020), including 10 days for dietary adaptation. Milk yield was recorded daily. Rumen fluid and milk samples were collected after 2 h of feeding in the morning of day 0, 15, 30, and 60. The data showed that rumen pH increased (*p* < 0.05) when cows were provided with SC. On average, the cows in the SC group produced 1.36 kg (*p* < 0.05) more milk per day than those in the CON group. Milk fat content of cows in the SC group was also higher (4.11% vs. 3.96%) (*p* < 0.05). Compared with the CON group, the concentration of acetic acid in the rumen fluid of dairy cows in the SC group was significantly higher (*p* < 0.05). There were no differences (*p* > 0.05) found in milk protein content and propionic acid between groups. The SC group had a tendency increase in butyric acid (*p* = 0.062) and total VFA (*p* = 0.058). The result showed that SC supplementation also enhanced the ratio between acetic and propionic. Most of the mean inflammatory cytokine (IL-2, IL-6, γ-IFN, and TNF-α) concentrations (*p* < 0.05) of the SC group were lower than CON group. This study demonstrated that high-yield cows receiving supplemental SC could produce more milk with higher fat content, have higher rumen acetate, and potentially less inflammatory cytokines.

## 1. Introduction

With the improvement of genetics, nutrition, and farm management, both the milk yield of dairy cows and the nutritional value of milk (e.g., protein and fat content) have been increased [1,2]. These improvements, however, pose threats to animal health and welfare due to metabolic stress. The cows, during the high-yielding period, are especially susceptible to metabolic and infectious diseases, such as digestive alterations, mastitis, rumen acidosis, and lameness [3]. To cope with the challenges, some additives were proposed to be added to the diet to enhance the productivity and health of animals, such as Boswellia sacra resin [4], rumen-protected fat [5], and *Saccharomyces cerevisiae* culture (SC) [6]. SC is a concentrated and dried product of *Saccharomyces cerevisiae* strain after solid or fluid fermentation. It contains fermentation substrates, bacterial proteins, yeast metabolites, yeast cell walls, and other beneficial substances, and it has the functions of balancing animal intestinal flora, improving immunity, relieving stress, and improving productivity [7]. Obeidat et al. [8] reported that adding SC to the diet increased the dry matter intake (DMI), nutrient digestibility, and rumen fluid pH of ruminants. It could also improve the milk quality of dairy cows [9]. Ogunade and McCoun [10] showed that the addition of SC could effectively stabilize the internal environment of rumen fermentation. A study revealed that SC increased the degradation of dry matter (DM) and neutral detergent fiber (NDF) of forage, which could be explained by the increased amount of total bacteria, fungi, protozoa, and lactate-utilizing bacteria in the rumen, and the decreased amount of starch-degrading and lactate-producing bacteria [11]. SC stabilized rumen pH, under normal conditions, tended to decrease ruminal lipopolysaccharide (LPS) after feeding, enhanced the milk fat depression during subacute ruminal acidosis (SARA), and reduced the extent of SARA-associated inflammatory cytokines [12].

In the previous studies, most of the SC experimental data were obtained under in vitro conditions, which mainly focus on ruminal microorganism fermentation [13,14,15,16], while some studies under in vivo situations paid more attention to milk yield instead of liver function and inflammatory cytokines [17,18,19]. Thus far, there is limited understanding of how SC would affect the performance of the high-yield cows associated with serum inflammatory cytokines, enzymes, metabolism, and rumen fermentation function. Therefore, this study systematically observed the effects of SC on the production performance, rumen pH, volatile fatty acid (VFA), serum biochemical, and immune indicators of high-yield dairy cows.

## 2. Materials and Methods

### 2.1. Animals, Diets, and Experimental Design

Forty healthy high-yield lactation Holstein cows (milk production > 30 kg/d) were selected and randomly divided into two homogeneous groups: parity (2.9 ± 0.7 vs. 2.7 ± 0.7), days of milk (73.2 ± 9 vs. 72.7 ± 6.8), weight (655.7 ± 38.1 kg vs. 656.1 ± 41.1 kg), body condition score (2.80 ± 0.1 vs. 2.86 ± 0.2), somatic cell counter (SCC, 23.6 ± 7.3 vs. 25.2 ± 8.8), fat percentage (4.02 ± 0.2 vs. 4.06 ± 0.3), protein percentage (3.26 ± 0.3 vs. 3.27 ± 0.3), lactose percentage (5.34 ± 0.1 vs. 5.37 ± 0.1), and daily milk production (40.8 ± 6.3 kg vs. 40.7 ± 5.2 kg). Body condition score (1–5 grading scale) was evaluated independently by 3 individual observers, and the average value was adopted for each cow, according to the method of Paul et al. [20]. The experimental group received a basal diet supplemented with the SC of 100 g/head/d (i.e., SC group) and control group fed the same basal diet without SC (i.e., CON group). The commercial name of SC is Naijiaoyi (lot number 201901050103 and patent number ZL2016200553625) provided by Xi’an Xinhanbao Biological Technology Company (Shanxi, China). The product contains the nutrients of the yeast cells (>1.5 × 10^11^/g), the metabolites formed after fermentation, and the denatured medium needed for the growth of lactic acid bacteria. The DM content of the product was 93%, and the content of crude protein, crude ash, and mannan was 16.9%, 3.9%, and 2.3%, respectively (as DM basis). The composition and nutrient levels of the diet are shown in Table 1. The nutritional requirements of dairy cows met the standard in NRC (2001) [21]. The free-stall barn housing system was adopted: feeding and milking three times a day, drinking water ad libitum. The experimental period lasted approximately 70 days (from 18 January to 27 March 2020), including 10 days for dietary adaptation. The trial was carried out in Beijing Shounong Animal Husbandry Development Company (39°30’ N, 116°33’ E) in northern China.

### 2.2. Data and Sample Collection

#### 2.2.1. Milk Yield and Milk Profile

Daily milk yield was recorded using an automatic milking system (ALPROTM by DeLaval©, Tumba, Sweden). Milk samples were collected after 2 h of feeding in the mornings of day 0, 15, 30, and 60 of the experiment period. Individual milk samples (500 mL/each) were collected according to the ratio of 4:3:3 for morning, afternoon, and evening, respectively. The obtained milk samples were added to the potassium dichromate preservative and stored at −4 °C refrigeration. The milk samples were sent to Beijing Dairy Cow Center for composition analysis, including milk protein, milk fat, lactose, SCC, and DM content. A near-infrared reflectance spectroscopy analyzer (Seris300 CombiFOSS; Foss Electric, Hillerød, Denmark) was used for determination, which is a seamless integration of MilkoScan^RM^ (Hillerød, Denmark) and Fossomatic^TM^ (Flow Cytometry, Hillerød, Denmark).

#### 2.2.2. Blood Sample Collection and Chemical Composition Determination

Blood samples were obtained from each animal at day 0, 15, 30, and 60 of the formal trial period. The samples were collected from the jugular vein in vacuum tubes without anticoagulants. Serum was obtained by centrifugation (3000× *g* for 15 min at 4 °C). An aliquot of each sample was frozen and stored at −20 °C for chemical analysis. The following biochemical blood components were measured by an auto-analyzer (CLS880, ZECEN Biotech Co., Ltd., Qingdao, Shandong, China): total bilirubin (TBIL), total protein (TP), albumin (ALB), alkaline phosphatase (ALP), alanine aminotransferase (ALT), glucose (GLU) and triglyceride (TG), in which TBIL, TP, ALB, ALP, and ALT were tested by kits from ZECEN Biotech Co., Ltd. (Jiangsu, China), and GLU and TG were tested by kits from Jiancheng Bioengineering Institute (Nanjing, China). β-hydroxybutyrate (BHB) and non-esterified fatty acid (NEFA) levels were measured using a spectrophotometer (Model 722, Gaomi Caihong Analytical Instrument Co., Weifang, Shandong, China) with kits supported by Jiancheng Bioengineering Institute (Nanjing, China). The levels of cytokines in the samples were measured by ELISA kit (ELISA, Thermo Multiskan Ascent, US), including interleukin-1β (IL-1β), interleukin-2 (IL-2), interleukin-6 (IL-6), interleukin-10 (IL-10), γ-IFN (γ- interferon), and tumor necrosis factor-α (TNF-α). The lowest detection level of the marker that the antibody pair used in the ELISA kit was < 5 pg/mL. The coefficients of variation of inter-assay and the intra-assay were 4.8% and 4.2%, respectively.

#### 2.2.3. Rumen Fluid Collection and Analysis

At day 0, 15, 30, and 60 of the formal trial period, 200 mL rumen fluid from each cow was collected through various stomach tubes, a method described by Shen et al. [22]. The rumen fluid was immediately filtered with four layers of gauze, and the filtrate was divided into two 50 mL centrifuge tubes. Once collected, one of the aliquots was used to measure pH with the use of sophisticated handheld pH meters (Starter 300; Ohaus Instruments Co. Ltd., Shanghai, China). Another one was stored at a −20 °C refrigerator and analyzed for VFA (acetic acid, propionic acid, butyric acid, and total VFA). The concentrations of VFA in rumen fluid were determined using a gas chromatograph (6890N; Agilent technologies, Avondale, PA, USA) equipped with a capillary column (HP-INNOWax 19091N-213, Agilent). More details regarding the method were described in Cao et al. [23].

### 2.3. Statistical Analysis

The SAS 9.4 statistical software (SAS institute, Carry, NC, USA) was used for statistical analysis. Milk yield and milk index data were analyzed by split-plot in time ANOVA; for repeated measures refer to Tesfaye and Hailu [24].

Fermentation data, including pH, acetate, propionate, butyrate, total VFA, lactate, the ratio between acetate and propionate (A:P), and all of the blood parameters data, were analyzed by using repeated measures data of MIXED procedure with model (1).
Yijk = μ + Di + Tj + (DT)ij+ Aik + εijk,(1)
where μ is the overall mean. Di is the fixed effect of treatment (i = 1–2). Tj is the fixed effect of sample collecting time (j = 1–4). (DT)ij is the fixed interaction effect of Dj and Tj. Aik is the random effect of the animal within Di, and εijk is the random error. Time was used as a repeated measure.

The results were expressed as least squares mean and standard error of mean. *p* < 0.05 indicates a significant difference. 0.05 < *p* < 0.1 means there is a tendency difference.

## 3. Results

### 3.1. Effects of SC on Milk Yield and Profile

Table 2 showed that the milk yield and 3.5% fat corrected milk (3.5% FCM) of the SC group were increased by 1.36 kg/d (*p* < 0.01) and 2.63 kg/d (*p* < 0.001), respectively, when compared with the CON group. The addition of SC significantly increased milk fat percentage (*p* < 0.01), lactose percentage (*p* < 0.05), and DM content (*p* < 0.01). However, supplementary SC supplementation did not significantly change the milk protein in milk (*p* > 0.05). The SCC in the SC group was lower than that of the CON group (*p* < 0.01).

### 3.2. Effects of SC on Ruminal pH and VFA

As shown in Figure 1a,b, the SC group had a greater pH (*p* < 0.05) and a higher concentration of acetic acid in the rumen fluid (*p* < 0.05), compared with those of the CON group. The addition of SC did not affect propionic acid (*p* > 0.05) (Figure 1c). The SC group tended to have more butyric acid (*p* = 0.062) and total VFA (*p* = 0.058) (Figure 1d,e). The concentration of individual VFAs (*p* < 0.05) and total VFA (*p* < 0.05) were affected by the sampling time. Figure 1f showed that SC supplementation also enhanced A:P, but the gap between the CON and SC group became smaller over time. Meanwhile, the A:P of the SC (from 2.39 to 2.66) and CON (from 1.94 to 2.64) group was also affected by the time of sampling (*p* < 0.05). There was no interaction effect found between the treatment and the time for the individual VFAs (*p* > 0.05), total VFA concentration (*p* > 0.05) and A:P (*p* > 0.05).

### 3.3. Effects of SC on Hepatic Function and Energy Metabolism

The serum TBIL (*p* < 0.01) and ALT (*p* < 0.01) decreased over time, but the values between the two groups did not differ (*p* > 0.05) (Table 3). The concentration of serum total protein (TP) (*p* < 0.01) was lower in SC group, compared with the control group (Table 3). It was not affected (*p* > 0.05) by the interaction of sampling time and treatment group (Table 3).

Table 4 shows that the sampling time affected the concentration of glucose (GLU) (*p* = 0.001). The treatment and interaction did not change the concentration of non-esterified fatty acid (NEFA) (*p* > 0.05), β-hydroxybutyrate (BHBA) (*p* > 0.05), GLU (*p* > 0.05), and triglyceride (TG) (*p* > 0.05).

### 3.4. Effects of SC on Inflammatory Cytokine

The effects of the treatment group on inflammatory cytokine concentrations are summarized in Table 5. Most of the mean inflammatory cytokine (IL-2, IL-6, γ-IFN, and TNF-α) concentrations (*p* < 0.05) in the SC group were lower than those in the CON group. The IL-1β concentration had a tendency decrease (*p* = 0.061), but the IL-10 was unaffected (*p* > 0.05). The effect of time and the interaction of time and treatment did not affect these inflammatory cytokine concentrations in blood (*p* > 0.05).

## 4. Discussion

A random-effects meta-analysis showed that the increase in milk production estimated for cows supplemented with SC in peer-reviewed studies was 1.2 kg/d more milk, 1.6 kg/d more 3.5% FCM, or 1.7 kg/d more energy-corrected milk [9]. A recent study also revealed that milk fat percentage and milk production were significantly increased by feeding SC to dairy ruminants [26]. In our study, the addition of SC significantly increased the milk production (1.36 kg/d) and 3.5% FCM yield (2.63 kg/d) of high-yield dairy cows. Our study showed that the percentage of milk fat was increased by 0.15% in SC supplementation cows. The increases in both milk production and milk fat percentage result in a higher milk fat yield. This is in line with the results in Poppy et al. [9] and Ma et al. [26]. These effects could be probably explained by Li et al. [12], which showed the stabilizing effect of SC on rumen pH and fermentation.

The variation of ruminal pH is related to many factors. One of the most important is the dietary type, which has impacts on the rumen fermentation model. Nowadays, high-yield lactating dairy cows are always fed with high proportion of rapidly fermentable non-fiber carbohydrates, which affect the ruminal pH stabilization [27]. In our study, the high-yield cows supplemented with SC had significantly higher rumen pH. This might be explained by the increasing of lactate-utilizing bacteria [28], which reduced lactate concentrations in the rumen [29]. It suggested that this SC product stabilized rumen acidity. The stabilized rumen condition allows enhanced growth and activity of fiber-digesting bacteria [30], resulting in improved fiber digestion [31] and, subsequently, higher acetic acid production and A:P ratio in the rumen [32]. Similar results were found in our study, the SC supplementation cows have a higher acetic acid production, and A:P ratio. Milk fat concentration is highly influenced by nutrition and rumen fermentation. For example, studies show that milk fat concentration is positively associated with acetate [33,34], which was also observed in our study. In the present research, the milk protein and propionate did not differ between SC and control group, which is in line with the results in Desnoyers et al. [35] and Thrune et al. [36], who found SC had no influence on milk protein and propionate concentration. In addition, the VFA production decreased over time in both groups (Figure 1). This might be partly explained by the high-concentrate diet used during the entire experiment period, with a concentrate to forage ratio of 60:40. This was similar to the study in that long-term high-grain diet feeding gradually lowers VFA production in cattle [37]. Further studies are required to better understand those variations.

This study showed that there were no differences between treatments in concentrations of TBIL, ALB, ALP, ALT, NEFA, BHBA, GLU, and TG in blood serum, in which the TBIL, ALB, ALP, and ALT relate to the liver function of the dairy cows. Therefore, our study suggested that the SC supplements did not affect liver function in high-yield lactating cows. The NEFA, BHBA, GLU, and TG reflect energy metabolism. Hence, it indicates that the SC supplementation did not alter the energy metabolism of the high-yield cows. These results were inconsistent with some other studies, who concluded that the live yeast supplementation favorably influenced the metabolic status and might have a liver-protecting effect on the high-yield cows [38,39]. The non-significant results in our study could probably be explained by the fact that the cows used in this experiment were at approximately 70 days into lactation. The negative energy balance is less pronounced at this time, once the peak of ingestion was reached. However, the data obtained in our study showed that the lactose content of milk increased in the SC group. Energy balance in dairy cows is positively correlated with lactose percentage [40,41], especially for high-yield cows [42]. It implies the SC has the potential to affect energy metabolism.

The concentration of serum TP decreased when the SC was added to the diet. Elevated TP may indicate inflammation or infections. In agreement with this finding, we observed that the proinflammatory cytokines (IL-2, IL-6, γ-IFN, and TNF-α) of dairy cows were significantly lower in the SC group compared with the CON group. A previous study showed that being fed with high concentrate diets is simultaneous with activation of a non-specific acute phase reaction (APR) in cows [43,44], triggering the activation of a systemic APR due to the translocation of LPS into the blood of systemic circulation stimulates the release of proinflammatory cytokines, such as IL-1, IL-6, and TNF-α by liver macrophages [45]. As a strong negative relationship between rumen pH and high concentrate diets was well known [46], we inferred that the decrease in proinflammatory cytokines (IL-2, IL-6, γ-IFN, and TNF-α) in SC group cows was due to the higher pH, which reduced the LPS flow into the systemic circulation. Previous studies also showed that feeding SC to the dairy cows in early lactation could reduce plasma LPS concentration and relieve the symptoms of sub-acute ruminal acidosis [47,48]. A previous study showed a significant negative relationship between pre-feeding rumen pH and concentration of LPS in the rumen fluid [49], and rumen pH play a modulatory role in the accumulation and release of LPS attribute to its effects on metabolic processes, changes in the cell membrane of rumen bacteria, maintenance of bacterial ecological balances, and other physiological functions of the rumen [50]. Our results showed that SC has the potential to reduce the inflammatory cytokines, which partially explained the lower SCC level in the milk of SC supplementation cows. This effect of SC may be partly attributed to the stabilizing effect on rumen fermentation, as mentioned above.

## 5. Conclusions

This study showed that supplementing high-yield lactating cows with 100 g/d SC increased milk production, milk fat content, and milk lactose content, but did not affect protein content. SC supplementation affected overall ruminal VFA concentration and induced a significantly greater ruminal pH of dairy cows. It had the potential to enhance the rumen microbial growth and decreased the inflammatory cytokines, but it did not affect the blood parameters reflecting liver function and energy metabolism. It is concluded that the SC supplementation could increase the performance and health of high-yield dairy cows by stabilizing the rumen environment and decreasing the inflammatory cytokines.

## Figures and Tables

**Figure 1 animals-11-02401-f001:**
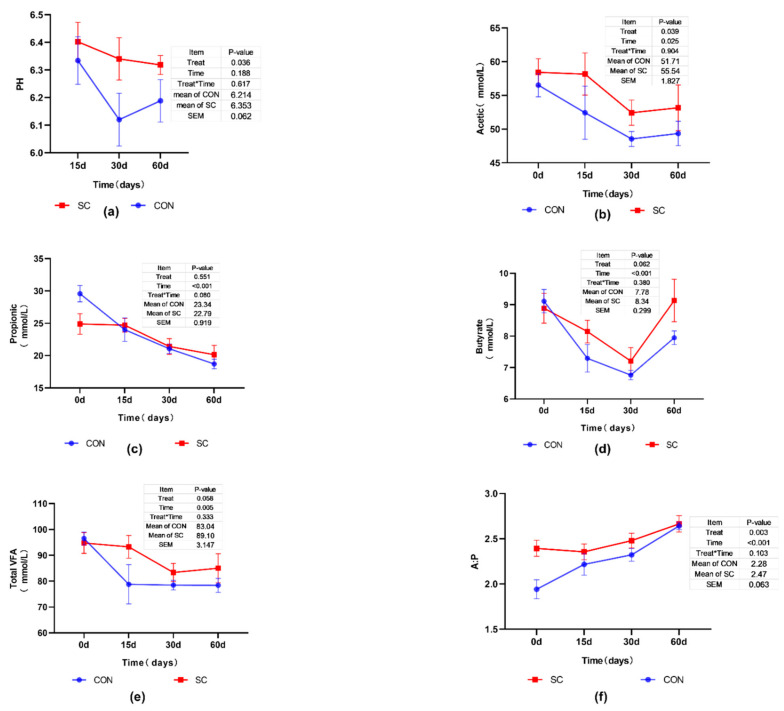
The effect of *Saccharomyces cerevisiae* culture (SC) on rumen VFA fermentation. SC is the experimental group diet supplemented with SC; CON is the control group without SC. (**a**–**f**) is the picture of pH, acetic, propionic, butyrate, total VFA concentrate, and acetate: propionate (A:P) with or without SC during the period, respectively. The data presented in the picture is mean ± standard error of mean.

**Table 1 animals-11-02401-t001:** Composition and nutrient levels of the basal diet (DM basis).

Items	Basal Diet (%)
Ingredients	
Whole corn silage	26.8
Alfalfa hay	12.5
Oat grass	1.6
Steamflaked corn	11.6
Extruded soybean meal	1.6
High yield concentrate ^1^	39.1
Soybean hull	1.2
Fat power	2.1
Beet pulp	1.1
Sunflower meal	0.8
Molasses cane	1.6
Total	100
Nutrient levels	
NE_L_/(Mcal/kg) ^2^	1.8
NDF	30.7
ADF	20.4
CP	17.8
EE	5.9
Starch	27.3
Calcium	1.0
Phosphorus	0.5

^1^ high yield concentrate is a concentrate supplement provided by Beijing Capital Agribusiness Group, including (DM basis): CP 24.75%, NDF 21.11%, Starch 36.26%, Fat 3.41%, Ash 7.54%, Calcium 1.04%, Phosphorus 0.53%, NaCl 1.0%, Fe 105 mg/kg, Zn 65 mg/kg, Mn 24 mg/kg, Cu 7 mg/kg, Mg 2 g/kg, K 10 g/kg, VA 20,000 IU/kg, VD 2300 IU/kg, and VE 88 IU/kg. ^2^ NE_L_: Net energy of lactation, a calculated value according to NRC (2001) [21], while the other nutrient levels were measured values.

**Table 2 animals-11-02401-t002:** Effects of SC on dry matter intake milk yield and milk composition of lactating dairy cows.

Items	CON ^3^	SC ^4^	*p*-Value
Milk yield (kg/d)	46.58 ± 0.36	47.94 ± 0.29	0.003
3.5% FCM (kg/d) ^1^	50.03 ± 0.57	52.66 ± 0.52	<0.001
Milk composition			
Milk fat (%)	3.96 ± 0.16	4.11 ± 0.15	0.005
Milk protein (%)	3.29 ± 0.0.06	3.23 ± 0.05	0.54
Milk lactose (%)	5.19 ± 0.03	5.29 ± 0.03	0.046
DM content (%)	13.14 ± 0.16	13.84 ± 0.19	0.006
SCC/(×10^4^/mL) ^2^	32.4 ± 1.37	23.4 ± 0.98	0.005

^1^ 3.5% FCM: 3.5% fat corrected milk = (kg milk × 0.432) + (kg fat × 16.216) [25]. ^2^ SCC: Somatic cell count. ^3^ CON: Control group. ^4^ SC: Treated group.

**Table 3 animals-11-02401-t003:** Effects of SC on hepatic function.

Item ^1^	CON ^2^	SC ^3^	SEM (±) ^4^	*p*-Value		
				Treatment	Time ^5^	Interaction ^6^
TBIL (umol/L)	7.75	7.18	1.68	0.185	<0.001	0.204
TP(g/L)	77.47	73.97	3.48	<0.001	0.088	0.575
ALB (g/L)	40.07	40.7	1.16	0.100	0.475	0.749
ALP (U/L)	50.75	55.95	13.88	0.225	0.227	0.167
ALT (U/L)	30.60	30.03	4.66	0.513	<0.001	0.014

^1^ TBIL = total bilirubin, TP = total protein, ALB = albumin, ALP = Alkaline phosphatase, and ALT = alanine aminotransferase. ^2^ CON: Control group. ^3^ SC: Treated group.^4^ SEM: Standard error of the mean. ^5^ Time: Sampling time effect. ^6^ Interaction: The interaction between sampling time and treatment group.

**Table 4 animals-11-02401-t004:** Effects of SC on energy metabolism of high-yield lactating cows.

Item ^1^	CON ^2^	SC ^3^	SEM (±) ^4^	*p*-Value		
				Treatment	Time ^5^	Interaction ^6^
NEFA (umol/L)	42.74	41.96	7.45	0.790	0.445	0.117
BHBA(mmol/L)	0.37	0.32	0.08	0.125	0.918	0.396
GLU (mmol/L)	4.25	4.34	0.50	0.493	0.001	0.568
TG (mmol/L)	0.25	0.27	0.04	0.253	0.772	0.296

^1^ NEFA: Non-esterified fatty acid, BHBA = β-hydroxybutyrate, GLU = glucose, and TG = triglyceride. ^2^ CON: Control group. ^3^ SC: Treated group. ^4^ SEM: Standard error of the mean.^5^ Time: Sampling time effect. ^6^ Interaction: The interaction between sampling time and treatment group.

**Table 5 animals-11-02401-t005:** Effects of SC on inflammatory cytokine of high-yield lactating cows.

Item ^1^	CON ^2^	SC ^3^	SEM (±) ^4^	*p*-Value		
				Treatment	Time ^5^	Interaction ^6^
IL-1β (ng/L)	44.51	39.60	7.62	0.061	0.820	0.487
IL-2 (pg/mL)	205.73	180.30	33.17	0.025	0.928	0.978
IL-6 (ng/L)	519.06	454.81	72.67	0.009	0.994	0.854
IL-10 (ng/L)	214.84	201.77	26.69	0.161	0.896	0.901
γ-IFN (pg/mL)	87.93	77.69	14.03	0.028	0.703	0.684
TNF-α (ng/L)	370.49	324.79	57.49	0.018	0.953	0.772

^1^ IL = interleukin, γ-IFN = γ- interferon, and TNF-α = tumor necrosis factor. ^2^ CON: Control group. ^3^ SC: Treated group. is the experimental group received a basal diet supplemented with 100g/head/day SC. ^4^ SEM: Standard error of the mean. ^5^ Time: Sampling time effect. ^6^ Interaction: The interaction between sampling time and treatment group.

## Data Availability

Not applicable.

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
