# Peer review of "Effects of Saccharomyces cerevisiae Culture on Ruminal Fermentation, Blood Metabolism, and Performance of High-Yield Dairy Cows"

_animals, 2021, doi:10.3390/ani11082401_

Round 1

Reviewer 1 Report

The present study investigated the effect of adding SC to cows in their mid- lactation on milk production, milk composition, rumen fermentation, and some blood metabolites related to energy balance and inflammation.  This study can be interesting for the readers, but many flaws and justifications should be mended and done before reconsidering the study for further processing.

Address 2 : please correct group to Group

Line 20 and 29: Volatile Fatty Acid, remove capitalization of each word

 Line 23: ruminal volatile fatty acid, please use the abbreviation VFA, and be sure that you use it along the manuscript after mentioning for first time.

Line 27:  High milk yield is always achieved at the expense of the health of dairy cows due to its microbial stress. It is not clear what do you mean by microbial stress, to me it is very strange expression????

Line 32 and 38: remove extra full stop.

Line 33: it will be clearer if you show the period of supplementation as a period time and show when it started and for how long lasted relative to the lactation stage.

Line 36: On average, the cows in the SC group produced 1.36 kg more 36 milk per day than those in the CON group (P<0.05), (P<0.05) should be removed to follow the variable, eg produced (P<0.05), please correct and do this for all variables.

 Line 37: Milk fat content of cows in the SC group was 37 also higher (4.11% vs. 3.96%). Please show the significance level.

Line 82: please give information about animals’ weight and body condition score.

The cows used in this study were in their 70 days of milk. It means that were in the mid-lactation. Usually, dairy animals suffer from high milk yield repercussions during early lactation first 60 days of lactation. So, it is not clear why the treatment started so late when the negative impacts of milk high production become less effective.

Line 86-90: please add information about the number of yeast cells per each gram of the product. Also, more information about the product name, lot number, company location (province and country) is required.

Table 1: the diet ingredients are free of salts, premix, and calcium salts, that do not meet the animals’ requirements from these elements. If these items are included in the high yield concentrate, please state that.

Line 120: no details about the liver function metabolites was shown, so I got surprised when find a table for such parameters. Also, why plasma and serum were separated what was used for what??????

Line 177-118: please, show how the milk composition and SCC were measured, with the aid of automatic instruments, for example milk scan, or by conventional laboratory assay methods.

Line 125-126: show more details about the kits used for blood samples determination such as sensitivity of the method and the linearity.

Line 127: details are needed for the method used for determining NH3 and pH in the rumen fluid.

Line 133: show details about the method used for VFA determination. GC-detector type and conditions.

Statistical analysis

It is not clear to me why milk yield and milk composition attributes were analyzed by one way ANOVA. However, repeated samples were collected and repeated analyses were done for milk composition as well as SCC.

Table 2. Please show the source of the used equation for calculating CMF (add a reference)

Figure 2: is there a missed point, day 0 values

Figure: please revise all legends and be sure that you show all details such as means, and SE. Also, why in figures SE is used, while in tables SD. Please, use one type of variation tests.

Figure 4 could be merged with figure 3. There is no need for a new figure with a long legend.

No data was shown for NH3 levels in rumen

For all tables use one wat for showing SE, either SE for each group as in table 2 or pooled SE as shown in the rest of tables. Again, be sure if you use SD or SE.

In all tables explain what CON and SC refer to.

Line 247: revise and correct the sentence milk fat yield was not determined in this study and it is different than milk fat percentage.

270-272: this speculation about the effect of climatic conditions on the fermentation pattern and samples content from VFA seems not suitable. This is simply because both groups of animals have kept under the same conditions, also no interaction between treatment and time was observed in your work as shown in figures 3 and 4. However, time has significance effect but it could not be attributed to the climate conditions alone. So, I suggest to remove this figure and find another explanation for your data.

Line 274-284: To me it is expected to find no effect of treatment on energy-yielding metabolites. This is because the cows used in this study were 70 days in milk. This period is not a critical period and is not characterized by the negative energy balance problems. I advise you to show that in your discussion.

Reviewer 2 Report

Several yeast probiotics have been tested in dairy cows. There is a meta-analysis published in 2012 using 61 studies. So, the topis is not specially novel. There are some descriptions or data to be provided or clarified before acceptance for publication.

Line 28. Could you explain the concept “microbial stress”?

Lines 86. Did you replace SC by other ingredient, or it was just an addition?

Lines 94-95. If the study was from January 2019 to March 2020 how it can last 70 days? Were the animals enrolled in the study as they were calving?  If yes, you should describe it, if not, you should correct the dates

Table 1. Are the ingredients on DM basis as stated in the Table description?

Line111. How did you measure DMI? It does not appear in Table 2 neither feed efficiency

Line 197. Not described in materials and methods

Reviewer 3 Report

My comments and suggestions are attached below

Round 2

Reviewer 1 Report

The article was significantly improved, please try to consider the following comments

Minor comments

Saccharomyces cerevisiae should be italic

It is good to show reference for the method of body condition score evaluation.

I am not sure if this instrument is used to determine SCC, a near-infrared reflectance spectroscopy analyzer

For cytokines, please provide the sensitivity of the method, inter and intra assays coefficients.

Line 293-295: again, fat yield was not shown in this study, please re-write this part.

There are many out of date cited references: 1991, 1998, 2000, …..

The following studies may help in updating the reference list:

Soltan, Y. A., Morsy, A. S., Hashem, N. M., Sallam, S. M. 2021. Boswellia sacra resin as a phytogenic feed supplement to enhance ruminal fermentation, milk yield, and metabolic energy status of early lactating goats. Animal Feed Science and Technology, 114963.

Hashem, N.M., EL-Zarkouny S.Z., 2017.  Metabolic attributes, milk production and ovarian activity of ewes supplemented with a soluble sugar or a protected fat as different energy sources during postpartum period. Annals of Animal Science, 17:229-240.
